# Chronic Use of Artificial Sweeteners: Pros and Cons

**DOI:** 10.3390/nu16183162

**Published:** 2024-09-19

**Authors:** Lydia Kossiva, Kostas Kakleas, Foteini Christodouli, Alexandra Soldatou, Spyridon Karanasios, Kyriaki Karavanaki

**Affiliations:** 1Diabetes and Metabolism Clinic, 2nd Department of Pediatrics, National and Kapodistrian University of Athens, “P&A Kyriakou” Children’s Hospital, 11527 Athens, Greece; lydiakossiva@hotmail.com (L.K.); foteini.christodouli@nhs.net (F.C.); asoldat@med.uoa.gr (A.S.); spyroskara97@yahoo.gr (S.K.); 21st Department of Pediatrics, National and Kapodistrian University of Athens, “Agia Sophia” Children’s Hospital, 11527 Athens, Greece; koskakl2@yahoo.gr

**Keywords:** non-nutritional sweeteners, artificial sweeteners, side effects, glycemic control, diabetes, obesity, nonalcoholic fatty liver disease, cancer, gut microbiota, favorable effects

## Abstract

Over the past few decades, the scientific community has been highly concerned about the obesity epidemic. Artificial sweeteners are compounds that mimic the sweet taste of sugar but have no calories or carbohydrates; hence, they are very popular among patients suffering from diabetes or obesity, aiming to achieve glycemic and/or weight control. There are four different types of sweeteners: artificial, natural, rare sugars, and polyols. Artificial and natural sweeteners are characterized as non-nutritional sweeteners (NNSs) since they do not contain calories. The extended use of sweeteners has been reported to have a favorable impact on body weight and glycemic control in patients with type 2 diabetes (T2DM) and on tooth decay prevention. However, there is concern regarding their side effects. Several studies have associated artificial sweeteners’ consumption with the development of insulin resistance, nonalcoholic fatty liver disease (NAFLD), gastrointestinal symptoms, and certain types of cancer. The present review focuses on the description of different types of sweeteners and the benefits and possible deleterious effects of the chronic consumption of NNSs on children’s health. Additionally, possible underlying mechanisms of the unfavorable effects of NNSs on human health are described.

## 1. Introduction

Sweeteners offer a sweet taste to food and can be classified as carbohydrate (caloric) and noncarbohydrate (noncaloric) sweeteners. Caloric sweeteners include sugar and sugar-alcohols such as erythritol, sorbitol, mannitol, xylitol, maltitol, lactitol, and reducing starch syrup [1]. Noncaloric artificial sweeteners or non-nutritional sweeteners (NNSs) are a heterogeneous group of compounds with different chemical structures, which are popular substitutes for added sugars in foods and beverages due to their low caloric content and sweetness [2]. Non-nutritional sweeteners (NNSs) can be classified into artificial sweeteners (ASs) when chemically produced in the laboratory, and natural sweeteners (NSs) when directly extracted from plants. ASs include aspartame, saccharin, sucralose, neotame, acesulfame-k, and advantame, and NSs are mainly represented by stevia, made from extracts of the intensely sweet plant *S. Rebaudiana*, but also include hoodia, agavis, and Luo Han Guo Monk fruit extracts [3].

Non-nutritional sweeteners (NNSs) were introduced into the market as food ingredients in the 19th century [4]. Since then, NNSs have become extremely popular as sugar substitutes for weight control and prevention of obesity, especially during the past three decades, when the worldwide incidence of childhood and adulthood obesity has increased dramatically. From 1975 to 2016, a tenfold increase in the number of obese children and adolescents, aged 5–19 years, worldwide was reported recently [5]. Specifically, using body mass index (BMI), obesity rates among children and adolescents globally increased from less than 1% (i.e., five million girls and six million boys) in 1975 to approximately 6% in girls (i.e., 50 million) and 8% in boys (i.e., 74 million) in 2016 [5]. Apart from the common use of sweeteners for weight loss, diabetes management, and the prevention of dental caries [6,7], they are also added in pharmaceutical and other healthcare products, such as toothpaste and food supplements [8]. 

A recent review noted that 4–18% of total carbonated beverage intake among children consists of artificially sweetened beverages [9]. Furthermore, children due to their smaller size and high intake of beverages, consume the highest quantity of artificial sweeteners relative to their body weight per day [10]. Consumption should be in accordance with the established permitted daily limit (acceptable daily intake—ADI). ADI is measured in mg per kg body weight and expresses the amount of sweetener that appears safe for daily consumption for life. Usually, it is 100 times lower than the dose that proved toxic in animal experiments [11]. The Food and Drug Administration (FDA) and the European Food Safety Authority (ESFA) have determined the ADI for different sweeteners [12] (Table 1). The daily intake of four NNSs (acesulfame-k, aspartame, saccharin, and sucralose) in a sample of Irish preschool children was within (17–31%) ADI limits [13]. A study from Argentina showed that 17% of schoolchildren consumed NNSs, and 1.5% exceeded the ADI for cyclamate included in soft drinks [14]. Due to the use of multiple artificial sweeteners in products, it is unlikely that the consumption of a single one exceeds the ADI, unless only one NNS is consumed. Thus, the EFSA concluded that the consumption of NNSs by individuals of this age group does not cause safety concerns [15].

Apart from the significant beneficial effects of ASs, great concerns about their possible side effects on human health have been raised since 1970, when a correlation between saccharin consumption and bladder cancer was first reported in animal studies [16]. Artificial sweeteners have also been associated with hepatotoxicity [17]. There are contradictory results on the relationship between AS consumption and the risk of developing T2DM [18]. Furthermore, studies have revealed that ASs may affect blood sugar levels and result in insulin resistance either via impairing the normal cephalic response to food, or through the modulation of insulin and incretin production [19,20]. Currently, there is rising scientific interest regarding the impact of artificial sweeteners consumption on the development of nonalcoholic fatty liver disease (NAFLD) [21], which occurs with increased frequency among individuals with obesity and potentially leads to liver cirrhosis and hepatocellular cancer. Concurrently, the safety and benefits of natural sweeteners, like stevia, are under research. 

The present review focuses on the latest scientific data on the potential beneficial effects and negative impact of non-nutritional sweeteners, such as artificial and natural sweeteners on human health. Since low-calorie sugars (rare sugars, such as xylitol) and sugar alcohols (polyols) are sugars containing calories and are not considered NNSs, they are out of the scope of the current report. Our first aim was to report data on recent research regarding the different types of sweeteners and the effect of the chronic consumption of NNSs by children and adolescents on body weight, glucose and lipid metabolism, and on cardiovascular, dental, and general health. Our second aim was to mention the possible underlying mechanisms of the related unfavorable metabolic effects of their chronic use on human health.

## 2. Methodology

A search of the literature was performed in Pub Med for articles regarding the different types of sweeteners and their effects on human health, focusing on children and adolescents, published between June 2004 and June 2024. Childhood was defined as age less than 18 years. The following keywords were used: non-nutritional sweeteners, artificial sweeteners, diabetes, obesity, nonalcoholic fatty liver disease, cancer, gut microbiota, benefits. Three reviewers (K.L., Ch.F., and K.S.) performed the electronic search. Whenever the three reviewers disagreed with one another on the inclusion of an article, a fourth reviewer was recruited to decide on inclusion (K.Kakl.). 

The initial search of articles based on all the abovementioned key words combined revealed no relevant articles. Consequently, we continued our search with various combinations of keywords. The use of the keywords artificial sweeteners, side effects, diabetes, and glycemic control yielded 292 articles, 38 of which were relevant, based on the title and/or the abstract, and 254 were irrelevant. The use of the keywords artificial sweeteners and nonalcoholic fatty liver disease yielded 46 articles, 23 of which were relevant. The use of the keywords artificial sweeteners, gut microbiota, obesity, and diabetes yielded 225 articles, 33 of which were eligible. The use of the keywords non-nutritive sweeteners and cancer yielded 50 articles, 37 of which were relevant, based on the title. Finally, the use of the keywords non-nutritive sweeteners and favorable effects on health yielded 9 articles, all of which were relevant. Finally, 141 studies were included in the present review.

## 3. Effect of Sweeteners on Health

### 3.1. Beneficial Effects of Artificial Sweeteners

#### 3.1.1. The Use of Artificial Sweeteners in the Prevention of Tooth Decay

The use of artificial sweeteners in foods, beverages, and personal care products (aspartame, saccharin, and sucralose) seems to protect against the development of caries and tooth decay due to their antimicrobial activity on dental bacteria [22,23]. Moreover, in children with permanent dentition, the use of toothpaste with fluoride and xylitol may be more effective in preventing tooth decay than that with fluoride only [24]. However, other studies recommended the avoidance of NNSs because of their negative effect on the surface morphology of tooth enamel [24,25]. In general, the use of artificial sweeteners may have a preventive effect on the development of caries, but there is a limited number of studies in children comparing artificial sweeteners to sugar-containing products [26].

#### 3.1.2. The Effect of Sweeteners on Weight Reduction and Obesity Management

The potential role of artificial sweeteners on weight reduction and obesity management for healthy individuals remains controversial. There are studies linking the consumption of artificial sweeteners with a positive effect on obesity markers among children and adolescents. For example, the substitution of sugar beverages with low-caloric ones in children resulted in reduced BMI and body weight [27]. The use of sucrose in children was associated with a greater BMI z score decrease than sucralose [28]. The impact of replacing sugar drinks with artificially sweetened beverages or water was found particularly beneficial in a group of adolescents with the highest BMI [29]. On the contrary, other studies have shown no effect or even increase in weight or BMI with the use of artificial sweeteners among children and adolescents [30,31,32]. A study with 3682 participants concluded that those who consumed drinks containing sweeteners had a 47% greater increase in BMI than those who did not [33]. Additionally, the World Health Organization (WHO), as well as the American Academy of Pediatrics (AAP), concluded that the substitution of sugar-containing beverages and foods with NNSs is unlikely to lead to substantial and statistically significant weight loss in children, but they recognize that the data are limited [34]. Also, a meta-analysis of six cohort studies and two randomized-control trials including 15,641 children showed contradictory results on the effect of NNSs on body weight, while there are very limited studies on prenatal or infant exposure to NNSs [35]. The inconsistency in the results of existing studies may be attributed to the presence of confounding factors, such as increased food intake to compensate for the use of low-calorie beverages and obesity, and study design issues, such as short duration and heterogeneity of the study populations [36,37]. 

Regarding the effect of ASs on adipogenesis, the results of various studies are also controversial. Thus, one study [38] found that acesulfame-k- and saccharin inhibited lipolysis in 3T3_l1 cells culture and stimulated adipogenesis, while another study [39] found that aspartame reduced adipocyte differentiation and lipid accumulation. Thus, the effect of ASs on adipogenesis has not been clarified yet. Regarding the effect of stevia on adipogenesis, no relative studies are available to our knowledge. 

#### 3.1.3. The Role of NNSs on the Prevention of Metabolic and Cardiovascular Diseases

The increased prevalence of metabolic diseases, including obesity and type 2 diabetes, cardiovascular disease, and cancer, has been strongly linked with the adoption of unhealthy dietary habits [40,41,42]. In the United States, unhealthy diet was highlighted as the leading risk factor for death and third leading risk factor for disability-adjusted life-years loss [43]. Since overweight and obesity are increasing, there are efforts to limit dietary overconsumption and restrict energy intake, with the ultimate goal of preventing obesity and its associated metabolic disorders [44].

The consumption of sugar-sweetened beverages has been shown to lead to increased energy intake and obesity, along with associated cardiometabolic complications [45]. Therefore, a common weight loss strategy to restrict energy intake is to replace sugars with artificial sweeteners. Despite the WHO-recommended free sugar intake being below 10% of total energy intake, a great proportion of the European population exceeds this threshold, particularly children [46].

However, artificial sweeteners may have a different impact on energy balance, and subsequent body weight, compared to natural sugars through underlying physiological processes including the gut microbiota, the reward system, and adipogenesis. In addition, the attainment of long-term weight control with the use of ASs remains controversial [47,48]. 

#### 3.1.4. The Role of Sweeteners in Prevention of Reactive Hypoglycemia in T2DM 

Sweet-tasting sugars, even prior to ingestion, trigger physiological responses related to the release of insulin or incretins (glucagon-like peptide-1 (GLP1) and gastric inhibitory polypeptide (GIP)), which are gastrointestinal hormones produced by the small intestine that promote insulin secretion. However, artificial sweeteners are not able to prepare the gastrointestinal (GI) tract for digestion and utilization of nutrients the same way as sugars do [49].

Several randomized control studies in healthy individuals have shown that there was no cephalic insulin response upon the tasting of aspartame or sucralose, while an early rise in insulin concentration was found when tasting glucose [50,51]. Furthermore, artificial sweeteners, compared to natural sugars, do not directly induce incretin secretion, as this appears nutrient-dependent [52,53]. Moreover, insulin secretion is stimulated after the binding of both natural sugars (glucose and fructose) and artificial or natural sweeteners (stevia, sucralose, aspartame, cyclamate, and saccharin) on sweet-taste receptors located on pancreatic β-cells, but sweeteners stimulate insulin secretion less [54].

Furthermore, short- and longer-term (12–16 weeks) studies showed no effect of either artificial or natural NNSs consumption on insulin levels in healthy, diabetic, overweight, or obese individuals [55,56,57,58]. Thus, NNSs use does not seem to be associated with post-meal reactive hypoglycemia. 

#### 3.1.5. The Role of AS in Flavor Enhancement

Sweet taste of the foods is mediated by special receptors, the three G protein-coupled receptors (T1R1, T1R2, T1R3), that are expressed in the oral cavity, the pancreas, the gastrointestinal tract, adipose tissue, and the brain. The activation of sweet receptors by artificial sweeteners results in insulin and incretin secretion and glucose absorption by the gastrointestinal tract. However, saccharin and acesulfame-k have a bitter aftertaste as they bind to bitter receptors as well. Another artificial sweetener, cyclamate, acts as an antagonist of bitter receptors, and its combination with saccharin may mask the bitter aftertaste of the latter [59,60].

Additionally, the consumption of artificial sweeteners may alter the taste perception and, thus, the taste preferences. Under these conditions, naturally sweet foods, such as fruits and vegetables, are less appealing. This could lead to decreased consumption of nutrient-rich foods in favor of nutrient-poor foods and a negative impact on overall health. Animal models suggested that NNSs intake may alter normal response to caloric intake resulting in overeating, because NNSs lead to a sweeter diet [61]. In children, the consumption of NNSs activates the sweet receptors with still-elusive results in taste preference, metabolic profile, and appetite [62]. 

## 4. Unfavorable Effects of Sweeteners

### 4.1. NNS Effect on Insulin Sensitivity in Healthy Individuals and in Patients with T1DM and T2DM 

The results of studies on the effect of NNSs on diabetic control are inconclusive. Prolonged use of artificial sweeteners may be associated with the development of insulin resistance and T2DM in healthy individuals [63] or the deterioration of glycemic control in patients with diabetes [64], which raises particular concern as they are broadly used by patients with diabetes. 

In healthy individuals, the long-term use of artificial sweeteners, such as saccharin, acesulfame-k, neotame, and sucralose, has raised concerns about the future development of T2DM, though more research is needed in larger studies [65]. Longitudinal studies in adults have shown that the long-term consumption of NNSs is associated with T2DM development [63,66]. In one study only in healthy individuals, sucralose was found to decrease blood glucose levels soon after consumption, but the effect was not confirmed in patients with T2DM [67]. The relative studies in children are very limited. In one crossover study, the administration of aspartame in preschool children and of saccharin in school children resulted in higher postprandial blood glucose levels, in comparison with sucrose [68]. Thus, as NNSs are included in many sweetened beverages which are favorable to children, further studies on the use of NNSs during childhood are necessary in order to clarify their role on metabolic and neurodevelopmental effects in this sensitive age group. 

Regarding the effect of chronic use of NNSs in patients with diabetes, the studies have contradictory results. In one study in patients with T2DM, the use of aspartame was found to be safe, but it did not ameliorate glycemic control. On the contrary, it might increase the levels of cortisol and reactive oxygen species, leading to insulin resistance, with long-term use [65]. The mechanism will be explained in the relative chapter. However, a meta-analysis by Lohner S et al. of nine randomized interventional control trials (RCTs) including 979 patients with T1DM or T2DM, comparing the use of NNSs for 4–10 months with sugar or placebo, reported that there is inconclusive evidence on their effect on glycemic control (HbA1c) and body weight, and they reported an occurrence of adverse events in patients with diabetes [69]. On the contrary, in a literature review of interventional trials by Nadolsky ZK et al. [70], on the effect of chronic replacement of sugar-sweetened beverages (SSBs) or meals with “diet” (zero-calorie) alternatives in patients with T2DM or obesity, a favorable effect on weight loss was reported, attributed to energy deficit and improvement of impaired glycemic control.

According to evidence-based studies on the use of artificial sweeteners on glycemic control in patients with diabetes, the results are contradictory. However, most studies concur that the use of natural sweeteners (stevia, hoodia, and agavis syrup) may reduce insulin resistance, increase insulin sensitivity, and improve glycemic control in patients with type 1 and type 2 diabetes [71,72,73]. Thus, natural sweeteners could be a better alternative to sugar for patients with diabetes, making it easier for them to adjust to a nonsugar diet. 

### 4.2. NNSs Use and Gut Microbiota 

The artificial sweetener consumption is linked to the disruption of the gut microbiota leading to conditions related to prediabetes [74,75]. This has led to second thoughts on the consumption of NNSs from patients with diabetes. The controversy regarding the benefits versus negative effects of NNSs on public health imposes questions that are still unanswered. 

### 4.3. NNS Use and Cancer 

Although artificial sweeteners are an integral part of modern nutrition, severe concerns about their possible unfavorable effects on human health have arisen. In the 1970s, experiments in mice showed a correlation of the consumption of saccharin with the occurrence of bladder cancer [76]. Artificial sweeteners have also been associated with other malignancies, such as leukemia, lymphoma, and multiple myeloma (in men) [77]. Another study on the effect of sweeteners on the diversification and morphology of the DNA of the kidney and intestinal cells concluded that cells were less diverse and more susceptible to higher concentrations of sweeteners, cells from the intestinal epithelium were more affected, while saccharin and sucralose caused greater DNA fragmentation in all cell lines compared with other tested sweeteners [78]. Large studies from Europe showed conflicting results, with some of them indicating that high artificial sweeteners use was linked with increased overall mortality, cancer risk, and coronary artery disease, while others did not confirm such an association [28,64,79,80].

### 4.4. Other Unfavorable Effects of NNSs

Furthermore, sweeteners possibly cause hepatotoxicity [81]; prolonged consumption in animal studies has resulted in hepatocellular injury [82]. In particular, in studies on mice, various histological changes have been reported in liver sections under the effect of aspartame and saccharin [83]. Another side effect of sweeteners is their association with NAFLD [21,84], which will be analyzed in detail as follows.

Another side effect of artificial sweeteners is flatulence [85], since they are partially absorbed. Several sweeteners act as laxatives, especially sorbitol and mannitol.

Certain sweeteners have also been shown to cause side effects such as headache, fatigue, mood disorders, and dizziness in patients suffering from malignancies, depression, multiple sclerosis, and systemic lupus erythematosus [86]. 

Apart from the effect of artificial sweeteners on metabolic disorders and weight gain, their consumption during pregnancy has been associated with preterm birth. Therefore, their use should be limited among pregnant women, especially in those with any form of diabetes or insulin resistance [87,88]. The use of natural sweeteners in pregnant women and in young children was initially considered as safe, if used within stated acceptable daily intakes (ADIs) [89]. However, that position is now under revision, due to the lack of adequate evidence on their long-term use in early childhood [90].

The various reported side effects from the use of specific sweeteners are shown in Table 2. The favorable and unfavorable effects of sweeteners are reported in Table 3.

## 5. Pathophysiology of Unfavorable Effects of Sweeteners

### 5.1. Artificial Sweeteners and Nonalcoholic Fatty Liver Disease

Nonalcoholic fatty liver disease (NAFLD) is a severe complication of obesity, which may present even in childhood. NAFLD is a condition where fat accumulates in the liver, due to the disruption of de novo lipogenesis (DNL), fatty acid β-oxidation, fatty acid uptake, and very-low-density lipoprotein (VLDL) synthesis and secretion mechanisms in the liver [91]. Experimental studies in mice have revealed the association between consumption of artificially sweetened beverages and the development of NAFLD [92]. Also, a recent study that included adults from the U.S. identified a relationship between the consumption of AS beverages and the risk of NAFLD development [93]. Furthermore, another paper that included four European studies showed that the consumption of low/no-calorie beverages is associated with NAFLD [94]. Inversely, a systematic review and meta-analysis of seven observational studies reported that there is no clear connection between the consumption of artificially sweetened beverages (ASBs) and NAFLD due to insufficient studies in human subjects [95]. 

Sucralose is the most studied NNS for possible deleterious effects on the liver. Sucralose exerts its effect through various mechanisms, such as the stimulation of hepatic proinflammatory cytokines, the promotion of hepatic lymphocytic infiltration, and the increased hepatic lipogenesis [96,97]. Additionally, sucralose activates the T1R3-ROS-ER stress-dependent pathway [98]. The activation of T1R3 generates reactive oxygen species and triggers lipolysis and endoplasmic reticulum (ER) stress in the liver. ER stress stimulates the production of lipid droplets and interferes with very-low-density lipoprotein (VLDL) metabolism, both enhancing VLDL delivery to hepatocytes and inhibiting VLDL synthesis and export from these cells, which in turn triggers intracellular triglyceride accumulation, which favors the development of NAFLD [99,100]. ER stress also promotes apoptosis and reduces autophagy in the liver, which is connected to the development of NAFLD in mice [101]. In addition, sucralose alters the composition of gut microbiota, thus promoting the production of bile acids that have a proinflammatory effect on hepatocytes [102]. Hence, in order to elucidate any possible association between NNSs and NFALD, longer-term prospective studies in children and adults with objective methods measuring the intake of sweeteners are needed.

Regarding natural sweeteners’ effects on adipogenesis, there are very limited previous studies. Kakleas et al. [21] reported that stevia and trehalose may have a protective effect on NAFLD. An experimental study in *db*/*db* mice hepatocytes showed that stevia and stevioside attenuated liver steatosis through the mechanism of PPARa-mediated lipophagy [103]. 

Thus, most previous studies showed that AS consumption is associated with the development of NAFLD, whereas natural sweeteners, such as stevia, may have a protective effect on NAFLD. Further studies are necessary to elucidate these findings.

### 5.2. Artificial Sweeteners and Insulin Resistance 

Prolonged use of artificial sweeteners by healthy individuals may be associated with the development of insulin resistance and T2DM [104,105] or the deterioration of glycemic control in patients with diabetes [64]. This seemingly paradoxical association between artificial sweeteners consumption and metabolic disorders has been epidemiologically observed and explained by the following hypotheses.

One hypothesis is based on the suppression of the cephalic phase of digestion by the sweet taste artificial sweeteners [106]. Consumption of artificial sweeteners by mice led to hyperglycemic responses to oral glucose load combined with decreased levels of circulating GLP-1. This was not observed when glucose was infused directly into the stomach, suggesting that altered glucose homeostasis is related to the response to the sweet taste [107]. 

A second hypothesis supports that consuming artificial sweetener affects the gut microbiota. Artificial sweeteners, such as saccharin, sucralose, aspartame and stevia, resist fermentation by oral bacteria and have bacteriostatic activity [108,109]. They have similar effects on the gut microbiota in both animals and humans [75], impairing digestion and glucose homeostasis.

The third hypothesis supports that sweet taste receptors, including T1R (taste receptor one) and a-gustducin, respond to both caloric sugars, such as sucrose and glucose, and to artificial sweeteners, such as sucralose and acesulfame-k [110,111]. These receptors are also found in the intestinal mucosal secretory L cells, which secrete the peptide GLP-1 [112] and promote insulin excretion. The effect of NNSs on insulin sensitivity can be explained by activation of taste receptor type 1 member 3 (T1R3) and extracellular signal-regulated kinase (ERK1/2) signaling pathway [113]. Studies in mice have revealed that sucralose, through the activation of T1R3, generates reactive oxygen species and triggers lipolysis and endoplasmic reticulum stress in the liver [114]. Endoplasmic reticulum stress results in increased production of proinflammatory cytokines, such as TNF-α and interleukin 6, which further exacerbate inflammation, increase cortisol levels, and promote insulin resistance via the disruption of the insulin signaling pathway [115,116]. Furthermore, studies in trophoblasts treated with aspartame have exhibited cessation of cell proliferation due to increased oxidative stress [117]. Activation of the ERK1/2 pathway in hepatic cells results in decreased expression of adiponectin and increased lipolysis [113]. Consequently, the released free fatty acids from lipolysis can promote the expression of inflammatory cytokines and provoke an inflammatory response, contributing to the development of insulin resistance [118]. 

Studies in mice also suggest that AS consumption affects the absorption of glucose from the intestinal lumen cells by increasing the expression of the glucose transporter SGLT1 (sodium-dependent glucose transporter isoform 1) and GLUT2 (apical glucose transporter 2) [119,120]. 

On the other hand, a possible favorable effect of natural sweeteners on glucose metabolism has been reported in animal studies [121]. Specifically, moderate intake of hoodia decreased insulin resistance and inflammatory markers levels [122]. Although the underlying mechanism is still unclear, studies suggest an antidiabetic effect of stevia [123]. In particular, the phenols contained in the leaves of stevia—about 91 mg/g—are the main contributors to the antihyperglycemic activity. Indeed, the leaves of stevia have an antioxidative action, with the highest benefit being found in rats with diabetes [124].

Since only the last two mechanisms have been evaluated in humans, and existing studies’ methodologies vary, further research is needed to determine underlying pathophysiological mechanisms.

Thus, from the above studies, it is concluded that artificial sweeteners use is associated with the development of insulin resistance and T2DM, while, oppositely, natural sweeteners seem to decrease insulin resistance and may be beneficial for chronic use by patients with diabetes.

### 5.3. Artificial Sweeteners, Gut Microbiota, and Obesity

The GI tract is inhabited by many microbial species, including bacteria, viruses, and fungi, which have been shown to affect the host’s growth, metabolic, and immunological status [125]. NNSs are metabolized by the gut microbiota and have significant effects on biological mechanisms. A study in mice has shown that the use of acesulfame-k reduced the levels of *Akkermansia muciniphilia,* leading to glucose intolerance [75]. Also, the same study showed that the use of aspartame induced glucose intolerance by altering the abundance of gut microbiota [75]. Another study in mice showed that the administration of neotame reduced the abundance of *Firmicutes* and increased the abundance of *Bacteroidetes* [126]. In actuality, it has been shown that in children with obesity, the levels of *Firmicutes* are increased and those of *Bacteroidetes* are reduced [127]. There are three hypotheses on the pathophysiological mechanisms that are possibly associated with the use of NNSs and obesity development: 1. Uncoupling sweet taste with calories by using NNSs may negatively affect energy balance, leading to obesity. 2. Increased NNSs consumption leads to changes in the gut microbiota, to a more obesogenic one. 3. It is possible that NNSs act directly on gut microbiota, thus affecting host defense mechanisms and triggering inflammatory processes, leading to metabolic dysregulation [119,128].

### 5.4. Artificial Sweeteners and Cancer

The relationship between the consumption of artificial sweeteners and the occurrence of cancer according to meta-analyses is inconsistent. A possible link between NNSs use and carcinogenesis was initially investigated in animal models. In the 1970s, the FDA banned cyclamate based on scientific data indicating that high doses of cyclamate and saccharin given to rodents increased the risk of developing bladder cancer [129]. The Joint Food and Agriculture Organization/World Health Organization Expert Committee on Food Additives (JECFA) conducted an independent risk assessment of aspartame consumption and cancer in June 2023. The JECFA concluded that the ingestion of aspartame is not associated with adverse effects based on existing animal and human studies; therefore, the committee reaffirmed their recommendations on acceptable daily intake [130]. No other artificial sweeteners were identified as potential cancerous triggers in animal studies.

Another cohort study conducted in France, called the NutriNet-Santé Study, found that the consumption of great amounts of artificial sweeteners may result in carcinogenesis more often compared to controls [80]. They also reported that adults who consumed acesulfame-k had a slightly higher risk of cancer overall than those who did not consume acesulfame-k [80]. Because different studies have implied that artificial sweeteners are associated with obesity, and obesity is subsequently associated with cancer, the NutriNet-Santé investigators also assessed the risk of associations between artificial sweetener intake and obesity-related cancers as a group. A slightly higher risk of obesity-related cancers was found in consumers of higher amounts of all artificial sweeteners compared to the risk of nonconsumers. Nevertheless, another study from Australia found no association between artificially sweetened beverage intake and the risk of obesity-related cancers [131].

Studies in specific population groups have also been inconsistent. A study found that the intake of artificially sweetened beverages was associated with an increased risk of kidney cancer in a US cohort of postmenopausal women [132], but no association was found in a European cohort of healthy adults [133]. An “umbrella review” (i.e., a review of systematic reviews or meta-analyses) found a weak association between intake of artificially sweetened beverages and any type of cancer, and especially with colorectal cancer, pancreatic cancer, gastrointestinal cancer, and cancer mortality [134,135]. 

No studies have associated the consumption of sucralose, neotame, and advantame with the future development of cancer [136]. 

The inconsistency in the studies searching for causality in the association between artificial sweeteners and cancer is mainly due to the study design limitations. For example, individuals examined in various studies differ in significant parameters, including the quality and quantity of artificial sweeteners’ consumption. Therefore, seeking evidence for a causal relationship necessitates the evaluation of evidence from multiple harmoniously designed studies and the description of a plausible underlying pathophysiological mechanism to account for the connection [137].

Inversely, there is a possibility that natural sweeteners may be used as therapeutic agents for cancer. Thus, Khaybullin et al. [138] reported in 2014 that isosteviol triazole conjugates could be used for cancer therapy. The authors reported that the conjugates reduced the proliferation of cancer cell lines. The above finding is interesting and needs to be evaluated by further studies. 

## 6. Recommendations

Based on the present literature and the recommendations by different pediatric societies and the European Union regulations we suggest the following [79,139,140]:The daily carbohydrate consumption should be less than 10% of the total caloric intake, following a balanced and healthy diet.Infants less than 2 years of age should avoid the consumption of sweeteners.Children who suffer from phenylketonuria must avoid the consumption of aspartame and neotame.Children with type 1 or type 2 diabetes or obesity may benefit from the use of non-nutritive sweeteners, but always as a part of a moderate and balanced diet and healthy lifestyle.The use of NNSs in children has been associated with reduced dental caries development.It is important for pediatricians and healthcare professionals to be adequately trained in the use of sweeteners, to be competent to advise parents and children on the appropriate sweetener selection based on its proprieties.More high-quality research is needed for the use of NNSs in childhood, focusing on the appropriate age of exposure and taste preferences, neurodevelopment, and impact on the microbiome and its association to obesity, metabolic syndrome, and diabetes.

## 7. Conclusions

The widespread use and popularity of NNSs in modern nutrition make it imperative for healthcare professionals to become familiar with their benefits and provide patients with individualized advice on the appropriate sweetener selection, based on its proprieties and expected benefits. Non-nutritive sweeteners have several beneficial effects such as facilitating weight loss (although their long-term effect on weight reduction is ambivalent), diabetes management, flavor enhancement, and prevention of tooth decay. On the other hand, they may induce several side effects on human health, such as hepatotoxicity and NAFLD, metabolic disorders, and carcinogenicity. However, natural sweeteners, such as stevia, may have a protective effect for the development of NAFLD and increase insulin sensitivity.

Especially for children, since there are not enough studies regarding the long-term safety of NNSs, they should not be offered to children younger than 2 years. Children with phenylketonuria must also avoid aspartame and neotame. Therefore, further research should be conducted in this field to ascertain the safety of their consumption and determine the maximum safe daily consumption limits.

## Figures and Tables

**Table 1 nutrients-16-03162-t001:** Artificial and natural sweeteners—characteristics.

Sweetener	Acceptable Daily Intake (ADI)		Relevant Sweetness
	FDA (mg/kg)	EFSA (mg/kg)	
Accesulfame-k	15	9	200×
Aspartame	50	40	160–220×
Neotame	0.3	2	7.000–13.000×
Saccharin	15	5	300×
Sucralose	5	15	600×
Stevia	4^3^	4^3^	300×
Cyclamates	Not approved	7	30–40×

FDA: Food and Drug Administration, EFSA: European Safety Authority.

**Table 2 nutrients-16-03162-t002:** The unfavorable health effects of specific sweeteners.

	Hepatotoxicity	Tumors	NAFLD	Insulin Sensitivity	Weight Increase/Reduction
Aspartame	Yes	No	Yes	reduced	↓
Acesulfame-k		Yes		reduced	↓
Sucralose		No			↓
Saccharin	Yes	No	Yes	reduced	↓
Neotame					↓
Stevia	No	No	No	increased	↓
Hoodia	No	No	No	increased	↓
Agavis syrup	No	No	No	increased	↓
Rare sugars (D-psicose)	No	No	No	increased	↓
Fructose			Yes	reduced	↑

↓ = reduced, ↑ = increased.

**Table 3 nutrients-16-03162-t003:** Health effects of artificial sweeteners.

Advantages	Disadvantages	Potential Risks
In diabetes mellitus In weight control Prevention of dental carries Flavor enhancement In reactive hypoglycemia	No nutritional value Not managing to crave for sweets Inability to boost metabolic response after high caloric intake	Malignancy (very high dose) Hypertriglyceridemia (fructose) GI symptoms Insulin resistance and T2DM Deterioration of glycemic control in diabetes Weight gain Hepatotoxicity/hepatocellular injury Headache, mood disorders

## Data Availability

No new data were created or analyzed in this study. Data sharing is not applicable to this article.

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
