# Peer review of "Chronic Use of Artificial Sweeteners: Pros and Cons"

_nutrients, 2024, doi:10.3390/nu16183162_

Round 1

Reviewer 1 Report

Comments and Suggestions for Authors

The authors have submitted the entitled manuscript "CHRONIC USE OF ARTIFICIAL SWEETENERS: PROS AND 2 CONS." Below are several items for the authors to address/consider (to strength this review manuscript). Importantly, the authors should provide details on how they reviewed the available literature regarding artificial sweeteners for this review. In other words, what search engines were used, what key words we used, how were articles included/excluded? It might be outside of the scope of this review to follow the exact criteria of PRISMA (https://www.prisma-statement.org/), as I believe this is a narrative review, but minimally, the authors should provide details on keywords/search engines used to better aid the readers understanding of how the authors come to their concludes from the literature.

Introduction:

-The first paragraph is one sentence, followed by what seems to be the remaining portion of the first paragraph... perhaps this is an oversight, and a simple correction. Please consider combing the first sentence of the introduction with the (currently) second paragraph to have a complete opening paragraph.

-The mention of a dramatic increase in childhood and adult obesity is a very important statement. Perhaps a quick note of the %increase over the three decade time frame would aid the reader to see just how dramatic this really is.

-(Page 1, line 30)Avoid the use of the word "proved." Perhaps ease this wording to say that "the sweeters have been shown..." & should there be a citation to back this up here? Or, is the [2] citation the citation here? This currently reads as those that [2] citation is for the second portion of the sentence... Please clarify.

-There are some general grammartic errors: For example, saying "health care products, such as toothpaste and food supplements" does not work here if there is no "and" between "medicines" and "health care products"... Please consider checking the grammar and phrasing of this manuscript (Grammarly is a great resource to use for this).

-Not sure it is appropriate to start the third paragraph with the "however." The authors should reword this. Also, it is best to re-refer to the direct focus of the topic when starting the new paragraph, meaning when the authors say "their" in the opening sentence of the third paragraph, the question that comes to mind here is "who is their?" It does reason that this is meant as AS, but this needs to be more clear. Please reuse the direct terms/abbreviations to replace the "indirect" wording (in this case... their).

-Same grammar issues in the third paragraph as noted above... i.e., "Other reported side effects are hepatotoxicity, metabolic 41 disorders, such as insulin resistance and type 2 diabetes (T2DM)".. Please correct this as well. Also, is insulin resistance a "side effect" (??) or the condition that results from the problem/instigator in this context?

-The third paragraph is a bit sporadic, as there is mention of side effects (conditions, such as T2DM), then mention of the FDA approved AS and NNSs... (which might go better in the second paragraph where the FDA was first mentioned), then, there is a mention of ADI (also, please consider spelling out the abbreviation when starting the sentence off, or rewording to not start with the abbreviation). Collectively, these are small issues that could be fixed with a quick re-evaluation of the outline of the introduction.

-the case could be better made for why this review is being done. Is this review being done to shed light on the pros and cons? This is not clearly stated in the end of the intro. Also, the word "recent" is vague and (considering the above-mentioned expansion of the methods used to conduct a literature search) this needs to be more clear of if recent here means just within 5 years... 10 years... two decades.. or what exactly. This is important to note to better understand the literature search methods used for this review.

General Comments on remaining sections:

-There are several areas that have randomly bold and underlined words. Please correct.

- Page 2, line 81... was BMI defined/abbreviated before? Looks to be in the sentence following the first time being mentioned.. which needs to be flipped. Please check for this and correct if this is found elsewhere. I think WHO is used and then spelled out elsewhere as well... So, check abbreviations used to be sure this is all correct.

-Where GIP and GLP1 previously abbreviated?

The tables are a bit confusing... For example, table 2 .... these AS that have a "yes" under tumors.... does this mean they cause tumors? Also, is this a cause and effect that has been consistently shown with empirical evidence or just that there are correlations here? This is context that should be discussed in the main text.

-The conclusion ends with two separate one-sentence paragraphs. Please fix this as generally paragraphs are 3-4 sentences.

Comments on the Quality of English Language

Generally, the quality of English is fine. There are several grammatical issues and errors throughout that need to be addressed. Please consider using Grammarly (or a similar platform) to correct.

Reviewer 2 Report

Comments and Suggestions for Authors

This article reviews the effects of artificial sweeteners/non-nutritional sweeteners on human health, taste, flavour and diet. The citation is comprehensive and solid, which is good reference for understanding the effects of this important kind of sweeteners. My suggestion is that the authors could provide some concrete suggestions or guidances for use of these sweeteners in diet (e.g., kind and dose) based on their review.

Comments on the Quality of English Language

Some minor English writing: 

page 2, line 61, S.Rebaudiana should be S. Rebaudiana (blank after dot and italic). 

The reason to use words in bold (Potential risks) in Table 3? Maybe Advantages Disadvantages should also be in bold.

Reviewer 3 Report

Comments and Suggestions for Authors

 CHRONIC USE OF ARTIFICIAL SWEETENERS: PROS AND CONS

This work reviews the effects and consequences of artificial sweeteners on different aspects of human health. The topic is of interest, and the text structure appropriate. However, many improvements should be made:

-        Line 12: “Artificial sweeteners are substances used to replace sugar in patients with obesity or diabetes”. Please, rephrase it. Artificial sweeteners are used to mimic sweet flavor in food in a generalized way, not just for individuals suffering from diabetes or obesity.

-        Line 13: “1.artificial, 2.natural, 3.rare sugars, and 4.polyols”. Please, remove the numbers. This is a very short list and the use of numbering makes more difficult the reading instead of helping.

-        Line 20-21: “effects on human health in different age groups”. Is that the real objective of this review? Maybe you should reconsider the objectives and describe them a little bit better.

-        Lines 25-26: “…(NNSs) had been introduced into the market as food ingredients in the 19th century”. The reference is lacking. I would use: “since 19th century” or “were introduced”.

-        Lines 27-28: you introduce the problem of obesity in one sentence without reference nor connection with the other ideas. Maybe, after “obesity has increased dramatically.” you may include one sentence connecting obesity with the use of NNSs; or rewrite the following sentence to connect both ideas.

-        Line 41: “Other reported side effects are hepatotoxicity, metabolic disorders, such as insulin resistance and type 2 diabetes (T2DM)”. “reported” has been previously used, maybe you may use “found”. After hepatotoxicity I would use “or”, cause you only number two, followed by an example.

-        Line 49: the correct abbreviation of milligrams is mg, not mgr

-        The last paragraphs of the introduction are a little bit confusing, please rewrite the last 3 paragraphs in one. First, you should mention the kind of NNSs and why polyols are other sugars are not included in that definition, and then, you should indicate the objective of the review.

-        Additionally, you include in table 1 data from EFSA, but in the text you only mention FDA. I think it would be of interest to report information from EFSA also in the text.

-        Line 136: “artificial sweeteners leads in insulin and incretin secretion”. Change for “lead to” or “result in”

-        Lines 151-154: “may be associated with the development of insulin resistance and T2DM or the deterioration of glycemic control”. Line 156: “it may increase the levels of cortisol and free radicals,”. But, why? A deeper discussion of aspects like this (hypothesis, potential mechanisms implicated, etc.) is lacking.

-        The appearance of table 2 and 3 must be improved.

General aspects:

-        In some parts you refer to age groups like children, in other parts to patients with some pathologies… It seems that you have taken just the evidence you prefer instead of following some specific criteria. I recommend, in order to avoid that impression, that you try always to compare with other populations (e.g. children vs adults, patients vs healthy individuals, etc.)

-        Although the design of the overall structure is good, when you read every section, some parts are not well-organized, which make difficult it to follow. I encourage the authors to carry out a deep and quiet reading and rewriting of different parts, with a clear idea of the message you want to transmit.

-        Pay attention to the use of abbreviations: NNSs or NNS (select one and use that one); AS appears few times (maybe this abbreviation is not needed).

-        The text is a little bit superficial. The references and reported data are ok, but it seems just a description of the result of other studies. I miss a deeper discussion about the reasons/causes of the reported data throughout the text. It is true that the majority of times there is not a clear cause, or the underlying mechanisms are unknown; but there are many hypotheses, and authors use to provide their opinion about that in their manuscripts. Section 4b could be an example of that because you have mentioned more about this, however section 4a is very poor in this aspect (as well as other parts in the text).  I encourage you to make an effort and try to discuss some of these aspects you just mention in your manuscript.

-        Conclusion section should be shortened and rewritten (summarizing main results or important aspects of the review) after a deep maturation of the current manuscript.

Comments on the Quality of English Language

The quality of English is not bad, some changes must be made but they are more related to the content of the text than to grammar.

Round 2

Reviewer 1 Report

Comments and Suggestions for Authors

Thank you to the authors for taking the time to address the reviewer comments. These were thorough and great additions. This is a strong manuscript. Well done.

Author Response

We would like to thank Reviewer #1 for his/her very important comments that have significantly improved our document.

Reviewer 3 Report

Comments and Suggestions for Authors

The authors have carried out all the suggested changes, thank you for the effort. Simply one thing: there was a little misunderstanding with my first comment:

“Line 12: “Artificial sweeteners are substances used to replace sugar in patients with obesity or diabetes”. Please, rephrase it. Artificial sweeteners are used to mimic sweet flavor in food in a generalized way, not just for individuals suffering from diabetes or obesity”.

I didn’t mean you write that sentence literally. So, please, change this part in the abstract. You may write something like this: “Artificial sweeteners are substances used to mimic sweet flavor in food, which allows replacing sugar and makes them of great interest for individuals suffering from diabetes or obesity”. Something like that, but there is no need to be exactly this sentence.

Author Response

We would like to thank the reviewer for this very valuable comment. We have changed the sentence to "Artificial sweeteners are compounds that mimic the sweet taste of sugar but have no calories nor carbohydrates,  hence are very popular among patients suffering from diabetes or obesity, aiming to achieve glycemic and/or weight control".
